# DIFFUSION WORLD MODELS

## ABSTRACT

World models constitute a powerful and versatile tool for decision-making. Through their ability to predict future states of the world, they can replace environments for safe and fast simulation, and/or be leveraged for search at decision time. Advances in generative modeling have led to the development of new world models, that operate in visual environments with challenging dynamics. However, recurrent methods lack visual fidelity, and autoregressive approaches scale poorly with visual complexity. Inspired by the recent success of diffusion models for image generation, we introduce Diffusion World Models (DWM), a new approach to world modeling that offers a favorable trade-off between speed and quality. Through qualitative and quantitative experiments in a 3D videogame, real-world motorway driving, and RL environments, we show that Diffusion World Models are an excellent choice for simulating visually complex worlds.

## 1 INTRODUCTION

Learning a model of how an environment works has become central in the development of competent agents. World models can be used to augment real-world experience, generating fictional trajectories that improve the sample-efficiency of reinforcement learning (RL) agents (Hafner et al., 2023; Wu et al., 2023). They also allow an agent to explicitly reason about the future consequences of its actions, enabling planning and search (Schrittwieser et al., 2020). In addition to agent-centric uses, these models could lead to entirely new kinds of experiences, such as in gaming (Kim et al., 2020).

While world models offer numerous possibilities for simulating environments and improving decision-making, developing accurate models for complex domains remains an open challenge. A world model should ideally satisfy many desiderata: consistency over extended periods of time, internalization of sophisticated dynamics, handling of visually challenging observations, and adaptation to new policies. These requirements situate the design of effective world models at the intersection of a large number of learning problems such as memory, reasoning, vision, and continual learning.

Current world models are built using the previous generation of image synthesis methods – autoencoders map image observations to a low-dimensional latent state, where recurrent or transformer architectures model the environment's transition dynamics (Hafner et al., 2023; Micheli et al., 2022). However, these approaches have several drawbacks: recurrent architectures have limited expressivity since they do not model the joint distribution over future latent states. On the other hand, autoregressive transformers do model the joint distribution, but become prohibitively slow in visually challenging environments, as more tokens are required to encode frames and the attention mechanism scales quadratically with sequence length.

This paper takes inspiration from work in generative modeling for images and videos, where diffusion models have emerged as the favored approach (Dhariwal & Nichol, 2021; Ho et al., 2022a). We show how diffusion models can be adapted to world modeling. Having navigated a large space of denoising objectives, architectures, conditioning mechanisms and sampling schemes, we introduce Diffusion World Models (DWM), a new approach to world modelling that is able to effectively condition on both previous observations and actions, while allowing accurate and fast sampling.

We evaluate DWM's ability both to learn complex worlds from offline datasets, and to train RL agents in imagination. Experiments are performed across several environments: Atari games (Bellemare et al., 2013), a first-person shooter video game (Pearce & Zhu, 2022), and real-world motorway driving (Santana & Hotz, 2016). Compared to existing world models DreamerV3 (Hafner et al., 2023) and IRIS (Micheli et al., 2022), DWM offers better visual fidelity at competitive sampling speeds.

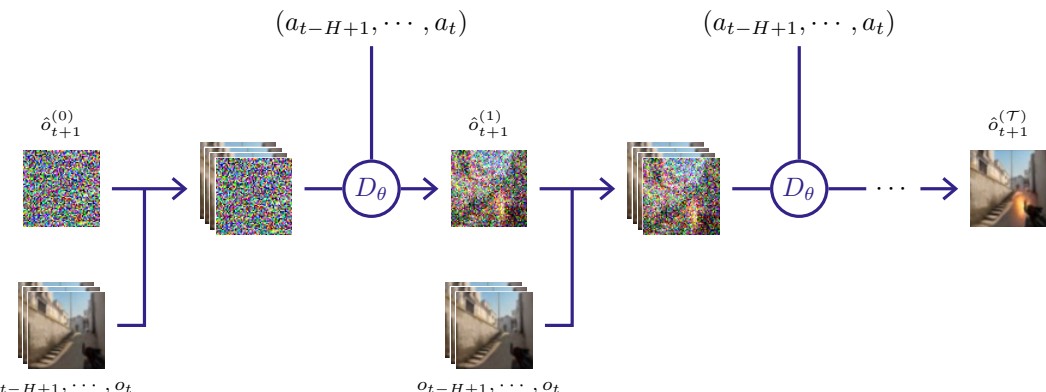

Figure 1: Overview of the diffusion world model (DWM) generating the next observation – the process starts from pure noise, $\hat{\mathbf{o}}_{t+1}^{(0)} \sim \mathcal{N}(0, \mathbf{I})$, and ends with a clean observation, $\hat{\mathbf{o}}_{t+1}^{(\mathcal{T})}$. At each denoising timestep $\tau = 1, \ldots, \mathcal{T}$, the model $D_\theta$ generates a less noisy observation given the current version, and is conditioned on previous observations $o_{t-H+1}, \ldots, o_t$ and actions $a_{t-H+1}, \ldots, a_t$. The generated next observation is coherent with the current scene, and reflects the last action $a_t$, here FIRE in the game CS:GO.

## 2 RELATED WORK

**World models.** Ha & Schmidhuber (2018) introduced the template for modern world models. They used a VAE to map image observations into a compact latent space, where the dynamics were modeled using an RNN. Successive generations of Dreamer agents improved on this approach, with recurrent state-space models (Hafner et al., 2020a), discretized and predictable latent representations (Hafner et al., 2020b), and more universal design choices (Hafner et al., 2023). A recent alternative approach trains a discrete autoencoder and autoregressive transformer. Dynamics learning is framed as a sequence modeling problem, with the transformer predicting a sequence of image tokens quantized by the autoencoder (Micheli et al., 2022; Robine et al., 2023). The present work continues the search for sound modeling choices, proposing diffusion models for the first time in this application.

**Image & video generation.** The last decade saw the leading technique shift from GANs (Goodfellow et al., 2014; Vondrick et al., 2016), to modeling individual pixels autoregressively using RNNs or CNNs (Van Den Oord et al., 2016; Van den Oord et al., 2016), to learning compressed latent representations with discrete autoencoders and modeling with autoregressive transformers (Esser et al., 2021; Yan et al., 2021), to diffusion models (Dhariwal & Nichol, 2021; Ho et al., 2022a). Our work shows how this final advancement, diffusion models, can also bring benefits to world models.

**Diffusion models.** These were introduced by Sohl-Dickstein et al. (2015), followed by several important refinements including a practical objective (Ho et al., 2020), faster samplers (Song et al., 2020a), and conditioning techniques (Ho & Salimans, 2022). Many large-scale models have centered around text conditioned images (Saharia et al., 2022; Rombach et al., 2022), or video (Ho et al., 2022a; Singer et al., 2022; Ramesh et al., 2022; Voleti et al., 2022). Most work on video diffusion models generates a set of consecutive frames simultaneously (Section 4) or by flexibly generating and conditioning on frames not in an autoregressive order Harvey et al. (2022). In contrast to this, a key requirement in our work is that frames *must* be generated autoregressively, as an agent's action should causally affect each subsequent frame. This necessitates innovation in our model's design.

**Diffusion & RL.** Janner et al. (2022) first showed that diffusion models could be effective in RL domains, by generating entire trajectories of states and actions. A large amount of follow up work has explored this further in environments with low-dimensional state spaces, e.g. (Ajay et al., 2022; Wang et al., 2022). Regarding image observations, Pearce et al. (2023) applied diffusion models to imitate human behavior, showing that diffusion could effectively *condition* on image observations. Lu et al. (2023) used diffusion models as a replacement for hand-crafted data augmentation schemes. They incorporated image observations by diffusing data in the low-dimensional latent space of a pretrained encoder. Our work is the first to use diffusion for efficient closed-loop interaction with an agent, as in model-based RL, and the first to diffuse pixels of image observations in RL settings.

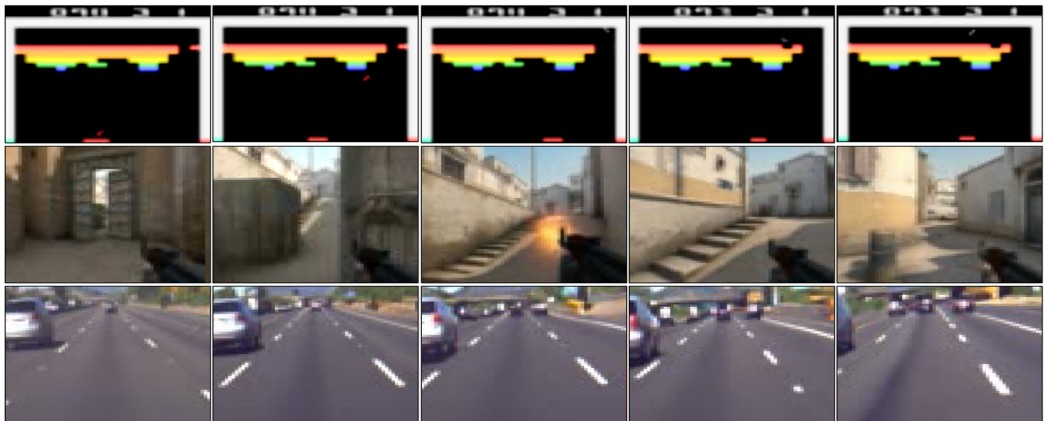

Figure 2: Example trajectories sampled from diffusion world models in the environments tested in this paper; 2D Atari games, a modern 3D first-person shooter, and real-world motorway driving.

# 3 BACKGROUND

## 3.1 WORLD MODELING

We consider a standard 7-tuple Partially Observable Markov Decision Process (POMDP), $(S, A, T, R, \Omega, O, \gamma)$, where $S$ is a set of states, $A$ a set of actions, $T$ the transition function describing $p(\mathbf{s}_{t+1} \mid \mathbf{s}_t, \mathbf{a}_t)$, $R : S \times A \to \mathbb{R}$ the reward function, $\Omega$ the set of observations, $O$ the observation probabilities conditioned on the state, $p(\mathbf{o}_{t+1}|\mathbf{s}_{t+1})$, and $\gamma \in [0, 1)$ is the discount factor.

We consider an agent acting in the environment in a closed loop. At each timestep $t$, it receives an image observation, $\mathbf{o}_t \in \mathbb{R}^{h \times w \times 3}$, and reward, $r_t$, and selects an action via its policy, $\mathbf{a}_t \sim \pi(\cdot)$, with $\mathbf{a}_t \in A$. Assuming access to observations but not states, the task of the world model is to predict $p(\mathbf{o}_{t+1}|\mathbf{o}_{t:t-H}, \mathbf{a}_{t:t-H})$, by implicitly inferring both the state transition function and the emission probabilities of observations. It conditions on a history of length $H$ to alleviate the partial observability of the environment.

There is ongoing debate about whether pixel-perfect predictions are required for world models. Hafner et al. (2020b) argue that they need *only* model task-relevant information, which necessarily leads to limited image predictions. In this work our goal is to learn *task-agnostic* world models that model the function, $p(\mathbf{o}_{t+1}|\mathbf{o}_{t:t-H}, \mathbf{a}_{t:t-H})$, at high visual fidelity. We believe that our goal will ultimately be useful in the pursuit of more general agents. As an example of this, some environments evaluated in Section 5 provide no explicit reward or discount factor, but this philosophy nevertheless allows us to train world models.

## 3.2 LIMITATIONS OF CURRENT WORLD MODELS

The design of current state-of-the-art world models have several limitations. The typical blueprint consists of two components. An autoencoder maps image observations to a low-dimensional, often discrete, latent state, $\mathrm{Enc} : \mathbf{o}_t \mapsto \mathbf{z}_t \in \mathcal{Z}$, where, $\mathcal{Z} = \{1, \dots, N\}^K$, is the set of K-tuples taken from a set of cardinality $N$. Recurrent or transformer architectures then model the environment's transition dynamics, $p(\hat{\mathbf{z}}_{t+1}|\mathbf{z}_t, \mathbf{a}_t)$. These can be decoded to image space, $\mathrm{Dec} : \hat{\mathbf{z}}_{t+1} \mapsto \hat{\mathbf{o}}_{t+1} \in \Omega$. Numerous popular works follow this template (Ha & Schmidhuber, 2018; Hafner et al., 2020b; Micheli et al., 2022), with various choices of autoencoder, dynamics model, and training objective.

**1) Latent bottleneck size.** The number of tokens in the latent representation, $K$, causes a fundamental trade-off – too low and it does not contain sufficient information to capture the visual complexity of the environment, too high and the latent dynamics models struggle and sampling time slows when modeling the joint distribution (Micheli et al., 2022). Diffusion world models avoid this issue by implicitly absorbing the encoder, decoder and dynamics models into a single architecture, that operates in pixel space without requiring a latent bottleneck.

**2) Latent sampling.** In order to model the joint distribution of the latent observation, autoregressive techniques can be used (Micheli et al., 2022), but this creates a sampling time complexity that's linear in the number of forward passes, $\mathcal{O}(K)$, for which the attention mechanism scales quadratically $\mathcal{O}((HK)^2)$. To alleviate this, some works sample latent dimensions simultaneously and *independently*, allowing for faster $\mathcal{O}(1)$ sampling (Ha & Schmidhuber, 2018; Hafner et al., 2020b; 2023), but at the cost of this coarse assumption. Methods like MaskGIT (Chang et al., 2022), falling in between autoregressive and independent sampling, are beginning to be explored (Yan et al., 2023). Diffusion world models allow sampling from the joint distribution in a way that depends on the number of denoising steps, $\mathcal{O}(\mathcal{T})$, rather than the number of tokens $K$, offering a more graceful way to trade-off sampling quality and speed, that is not baked in the model's architecture.

**3) Complexity.** Having two components requires either two separate training procedures (Micheli et al., 2022), or end-to-end optimization with multi-objective losses and backpropagation of gradients through sampling (Hafner et al., 2023). In contrast to these, diffusion world models offer a single end-to-end architecture using only a standard MSE diffusion loss.

## 4 METHOD

### 4.1 DIFFUSION MODELS

Diffusion models (Sohl-Dickstein et al., 2015) are a class of generative models inspired by non-equilibrium thermodynamics that learn to generate samples by reversing a noising process. While several flavors of diffusion models have been proposed (Ho et al., 2020; Song et al., 2020a;b), our work uses the formulation proposed in Karras et al. (2022), described below, which we found more robust than classical DDPM (Ho et al., 2020).

Assuming access to samples from a data distribution, $y \sim p(y)$, the goal is to learn to recover these samples from degraded versions, $x = y + \sigma\epsilon$, where, $\epsilon \sim \mathcal{N}(\mathbf{0}, \mathbf{I})$ is Gaussian noise, and $\sigma \in \mathbb{R}$ controls the level of degradation, sampled from some distribution, $\sigma \sim p_{\text{noise}}(\sigma)$. Concretely, we train a model, $D_\theta$, with the denoising objective,

$$\mathcal{L}(\theta) = \mathbb{E}_{y,\epsilon,\sigma} \left[ \|D_\theta(y + \sigma\epsilon, \sigma) - y\|^2 \right]. \tag{1}$$

Karras et al. (2022) parameterize $D_\theta$ as the weighted sum of the degraded input and the prediction of a neural network, $F_\theta$,

$$D_\theta(x, \sigma) = c_{\text{skip}}(\sigma)\, x + c_{\text{out}}(\sigma)\, F_\theta\big(c_{\text{in}}(\sigma)\, x, \sigma\big). \tag{2}$$

The conditioners, $c_{\text{in}}, c_{\text{out}} : \mathbb{R} \to \mathbb{R}$, are selected to keep the network's input and output at unit variance for a given noise level $\sigma$, and $c_{\text{skip}} : \mathbb{R} \to \mathbb{R}$ is given in terms of $\sigma$ and the standard deviation of the data distribution, $\sigma_{\text{data}}$, $c_{skip}(\sigma) = \sigma_{data}^2/(\sigma_{data}^2 + \sigma^2)$.

Combining these equations provides insight into the training objective of $F_\theta$,

$$\mathcal{L}(\theta) = \mathbb{E}_{y,\epsilon,\sigma} \left[ \| \underbrace{F_\theta\big(c_{\text{in}}(\sigma)\,(y + \sigma\epsilon), \sigma\big)}_{\text{Neural network prediction}} - \underbrace{\frac{1}{c_{\text{out}}(\sigma)}\Big(y - \frac{\sigma_{\text{data}}^2}{\sigma_{\text{data}}^2 + \sigma^2}\,(y + \sigma\epsilon)\Big)}_{\text{Neural network training target}} \|^2 \right]. \tag{3}$$

The network training target adaptively mixes signal and noise depending on the degradation level, $\sigma$. When $\sigma \gg \sigma_{\text{data}}$, we have, $c_{\text{skip}}(\sigma) \approx 0$, and the training target for $F_\theta$ is dominated by the clean signal, $y$. Conversely, when the noise level is low, $\sigma \approx 0$, we have, $c_{\text{skip}}(\sigma) \approx 1$, and the target is dominated by the noise, $\epsilon$. This approach yields a more robust training objective than DDPM, where the model *always* has to estimate the noise, which is trivial when the level of noise is high.

While the above description formulates the diffusion process in terms of a continuous value $\sigma$, at sampling time a discrete schedule $\sigma(\tau)$ is used, with $\sigma(\mathcal{T}) > \cdots > \sigma(2) > \sigma(1)$.

### 4.2 DIFFUSION WORLD MODELS

To extend the unconditional generative model of Section 4.1 to world modeling, we need to condition the denoising model, $D_\theta$, on a window of past observations and actions, as illustrated in Figure 1 – ultimately the neural network takes the form, $F_\theta\big(c_{\text{in}}(\sigma)\,(\mathbf{o}_{t+1} + \sigma\epsilon), \sigma, \mathbf{o}_{t:t-H}, \mathbf{a}_{t:t-H}\big)$. While

the requirements of such a model overlaps with those in the video generation literature, there are important differences in model functionality introduced by the world modeling scope that require several design changes compared to typical video diffusion.

**Requirement 1 – Generation must be autoregressive.** To be used *interactively* with an agent, a world model must autoregressively generate the next observation, with a causal dependency on each selected action. This differs from video generation, where frames may be generated simultaneously (Ho et al., 2022b), in blocks (Voleti et al., 2022), or in an arbitrary order (Harvey et al., 2022).

**Requirement 2 – Each generated frame conditions on an action.** In video generation, usually a single conditioning variable is common to all frames of a video, such as a text description. In world modeling, a new action is provided at each time step.

**Requirement 3 – Sampling speed should be minimized.** Whilst fast sampling is desirable for video generation, it is even more critical in world modeling, as it bottlenecks how fast agents train in imagination. Real-time generation is also a prerequisite for new user experiences (Kim et al., 2020).

### 4.2.1 ARCHITECTURE

The first of these requirements, autoregressive generation, requires conditioning on $H$ previous frames. We considered two architectures to achieve this – summarized in Figure 3.

**Frame-stacking.** The simplest way to condition on previous observations is by concatenating the previous $H$ frames together with the next noised frame, $\mathrm{cat}[\hat{\mathbf{o}}_{t+1}^{(\tau)}, \mathbf{o}_t, \ldots, \mathbf{o}_{t-H+1}]$, which is compatible with a standard U-Net (Ronneberger et al., 2015). This architecture is particularly attractive due to its lightweight construction, requiring minimal additional parameters and compute compared to typical image diffusion.

**Cross-attention.** The U-Net 3D (Ho et al., 2022b)[1] is a leading architecture in video diffusion (also in Figure 3). This inspired us to design and test a cross-attention architecture, formed of a core U-Net, that only receives a single noised frame as direct input, but which cross-attends to the activations of a separate history encoder network. This encoder is a lightweight version of the U-Net architecture – parameters are shared for all $H$ encoders, and each receives the relative environment timestep embedding as input. The final design differs from the U-Net 3D which diffuses all frames jointly, shares parameters across networks, and uses self-, rather than cross-, attention.

### 4.2.2 ACTION CONDITIONING & SAMPLING

In addition to observations, world models must condition on actions. After being embedded through a linear layer, these are concatenated or summed with the embedding of the noise level, $\sigma$, and input to the network through adaptive group normalization (AdaGN) Dhariwal & Nichol (2021).

Similar to text-to-image models, at training we optionally mask out the observation and action inputs with some probability (typically 0.2). By learning to generate observations with masked actions, the model must implicitly learn the data-generating policy. Once trained, the diffusion models are capable of being run both conditionally as a world model, and also unconditionally in video generation mode.

Training a diffusion model with masked actions also allows the option of using classifier-free guidance (CFG) (Ho & Salimans, 2022) when sampling. CFG has been a core innovation for text-conditioned generations, and we hypothesized that it might also be useful to focus the generations of DWM on the specific input action sequence. CFG works by combining conditional and unconditional network predictions in a weighted sum, for some CFG weight $w \geq 0$,

$$(1+w)F_\theta\big(\hat{\mathbf{o}}_{t+1}^\tau, \sigma, \mathbf{o}_{t:t-H}, \mathbf{a}_{t:t-H}\big) - wF_\theta\big(\hat{\mathbf{o}}_{t+1}^\tau, \sigma, \mathbf{o}_{t:t-H}\big). \tag{4}$$

When $w = 0$, vanilla conditional generation is performed, and $w > 0$ up-weights the conditioning.

## 4.3 LEARNING IN IMAGINATION

World models are frequently used to improve sample efficiency of RL methods, where an agent is trained in the imagination of a world model using the Dyna framework (Sutton, 1991).

---

[1] https://github.com/lucidrains/video-diffusion-pytorch

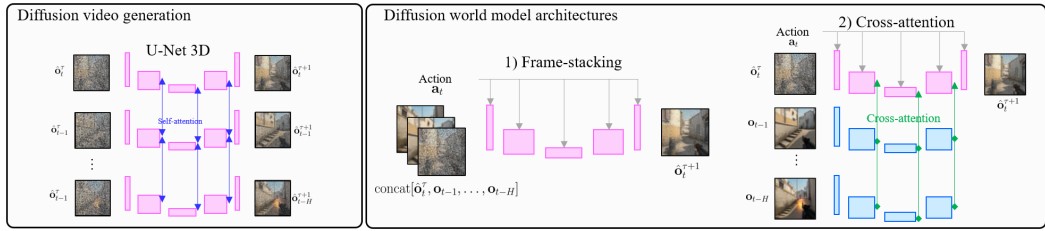

Figure 3: We tested two architectures for diffusion world models which condition on previous image observations in different ways. To illustrate differences with typical video generation models, we also visualize a U-Net 3D which diffuses a block of frames simultaneously.

**Step 1.** The RL agent collects experience in the real environment.
**Step 2.** This real experience is used to improve the world model.
**Step 3.** The RL agent is improved using fictional trajectories generated by the world model.
**Step 4.** Return to step 1.

We largely follow the RL setup used in a recent leading world model, IRIS: the actor-critic network is parameterized by a shared CNN-LSTM with policy and value heads, and the training objectives are REINFORCE with value baseline for the policy, and the Bellman error with $\lambda$-target for the value function. Full details can be found in Micheli et al. (2022).

## 5 EXPERIMENTS

Our experiments investigate the below questions.

**Q1) How effective is DWM at improving sample efficiency for RL agents?** We study this in a subset of Atari games, comparing to recent world models performing strongly in this domain.
**Q2) Do diffusion world models scale to more complex environments?** To understand the generality of DWM beyond 2D Atari games, we train it to model a modern 3D game, and in real-world driving.
**Q3) Can diffusion world models learn from offline and online data?** In the complex environments, DWM learns from fixed offline datasets, while int Atari it uses online data collection.
**Q4) How does visual quality and sampling speed compare to other world models?** As well as qualitative evaluations, we apply metrics from the video generation literature to evaluate the visual quality of imagined trajectories. We further report the sampling speed of various world models.
**Q6) How do critical hyperparameters affect performance of diffusion world models?** We ablate several key hyperparameters including architecture, number of sampling steps, and CFG.

### 5.1 WORLD MODELS OF COMPLEX 3D ENVIRONMENTS

#### 5.1.1 ENVIRONMENTS, BASELINES, METRICS

Our later Atari experiments will evaluate the utility of DWM in simpler environments for the purpose of training RL agents, trained online – a classic use of world models. In contrast, this set of experiments explores the value of DWM in modeling visually *complex* environments, by directly evaluating the visual quality of the trajectories they generate. We use two environments.

**CS:GO environment.** 'Counter-Strike: Global Offensive' is one of the world's most popular video games in player and spectator numbers. It was introduced as an environment for RL research by Pearce & Zhu (2022)[2]. We use the 3.3 hour dataset (190k frames) of high-skill human gameplay, captured on the 'dust_2' map, which contains observations and actions (mouse and keyboard) captured at 16Hz. We use 2.6 hours (150k frames) for training and 0.7 hours (40k frames) for evaluation. We resize observations to 64×64 pixels. We use no augmentation.

---

[2]https://github.com/TeaPearce/Counter-Strike_Behavioural_Cloning

Table 1: Results for 3D environments. Metrics are borrowed from the video generation literature. We compare real trajectories with generated trajectories of image observations, conditioned on a real sequence of actions, and an initial set of $H = 6$ observations.

| Method | CS:GO | | | Driving | | | Sample rate (Hz) ↑ | Parameters (#) |
|---|---|---|---|---|---|---|---|---|
| | FID ↓ | FVD ↓ | LPIPS ↓ | FID ↓ | FVD ↓ | LPIPS ↓ | | |
| DreamerV3 | 106.8 | 509.1 | 0.173 | 167.5 | 733.7 | 0.160 | 266.7 | 181M |
| IRIS ($K = 16$) | 24.5 | 110.1 | 0.129 | 51.4 | 368.7 | 0.188 | 4.2 | 123M |
| IRIS ($K = 64$) | 22.8 | 85.7 | 0.116 | 44.3 | 276.9 | 0.148 | 1.5 | 111M |
| DWM frame-stack (ours) | 9.6 | 34.8 | 0.107 | 16.7 | 80.3 | 0.058 | 7.4 | 122M |
| DWM cross-attention (ours) | 11.6 | 81.4 | 0.125 | 35.2 | 299.9 | 0.119 | 2.5 | 184M |

**Motorway driving environment.** We use the dataset from Santana & Hotz (2016)[3], which contains camera and metadata captured from human drivers on US motorways. We select only trajectories captured in daylight, and exclude the first and last 5 minutes of each trajectory (typically traveling to/from a motorway), leaving 4.4 hours of data. We use five trajectories for training (3.6 hours) and two for testing (0.8 hours). We downsample the dataset to 10Hz, resize observations to 64×64, and for actions use the (normalized) steering angle and acceleration. During training, we apply data augmentation of shift & scale, contrast, brightness, and saturation, and mirroring.

**Metrics.** To evaluate the visual quality and temporal consistency of generated trajectories, we use Fréchet Video Distance (**FVD**) (Unterthiner et al., 2018) as implemented by Skorokhodov et al. (2022)[4]. This is computed between 1024 real videos (taken from the test set), and 1024 generated videos, each 16 frames long (1-2 seconds). Models condition on $H = 6$ previous real frames, and the real action sequence. On this same data, we also report the Fréchet Inception Distance (**FID**) (Heusel et al., 2017), which measures the visual quality of individual observations, ignoring the temporal dimension. For these same sets of videos, we also compute the **LPIPS** loss (Zhang et al., 2018) between each *pair* of real/generated observations (Yan et al., 2023). **Sampling rate** describes the number of observations that can be generated, *in sequence*, by a single A6000 GPU, per second.

**Baselines.** We compare against two state-of-the-art world model methods; DreamerV3 (Hafner et al., 2023) and IRIS (Micheli et al., 2022), adapting the original implementations to our offline set up. We ensured baselines used a similar number of parameters to DWM. Two variants of IRIS are reported; image observations are discretized into $K = 16$ tokens (as used in the original work), or into $K = 64$ tokens (achieved with one less down/up-sampling layer in the autoencoder), which provide the potential for modeling higher-fidelity visuals.

**Compute.** All models (baselines and DWM's) were trained for 120k updates with a batchsize of 64, on up to 4×A6000 GPUs. Each training run took between 1-2 days.

### 5.1.2 RESULTS

Table 1 reports metrics on the visual quality of generated trajectories, along with sampling speeds, for the frame-stack and cross-attention DWM architectures, compared to baseline methods. DWM outperforms the baselines across all visual quality metrics. This validates the results seen in the wider video generation literature, where diffusion models currently lead (Section 2). The simpler frame-stacking architecture performs better than cross-attention, something surprising given the prevalence of cross-attention in the video generation literature – we believe the inductive bias provided by directly feeding in the input, frame-wise, may be well suited to autoregressive generation.

In terms of sampling speed, DWM frame-stac (with sampling steps $\mathcal{T} = 20$) is faster than IRIS ($K = 16$). IRIS suffers a further 2.8× slow down for the $K = 64$ version, verifying that sample time for autoregressive models is determined by the number of tokens $K$. On the other hand, DreamerV3 sampling speed is an order of magnitude faster – this derives from its independent, rather than joint, sampling procedure, and the flip-side of this is the low visual quality of its trajectories.

---

[3] https://github.com/commaai/research
[4] https://github.com/universome/stylegan-v

Table 2: Returns and human normalized scores at 100k frames for IRIS, DreamerV3, and DWM.

| Game | Return | | | | Human normalized score | | |
|------|--------|------|-----------|-------------|------|-----------|-------------|
| | Human | IRIS | DreamerV3 | DWM (ours) | IRIS | DreamerV3 | DWM (ours) |
| Asterix | 8503.3 | 854 | 932 | **4200.6** | 0.078 | 0.087 | **0.481** |
| Boxing | 12.1 | 70 | **78** | 1.7 | 5.825 | **6.491** | 0.133 |
| Breakout | 30.5 | **84** | 31 | 71.1 | **2.858** | 1.018 | 2.410 |
| Kangaroo | 3035.0 | 838 | **4098** | 2510.4 | 0.263 | **1.357** | 0.824 |
| Krull | 2665.5 | 6616 | **7782** | 4234.2 | 4.700 | **5.792** | 2.470 |
| Qbert | 13455.0 | 746 | **3405** | 2721.7 | 0.0438 | **0.244** | 0.192 |
| RoadRunner | 7845.0 | 9615 | 15565 | **20357.0** | 1.226 | 1.986 | **2.598** |
| UpNDown | 11693.2 | 3546.2 | N/A | 3198.9 | 0.270 | N/A | 0.239 |

Figure 2 shows selected examples of the trajectories produced by DWM in CS:GO and motorway driving. The trajectories are plausible, often even at time-horizons of reasonable length. In CS:GO, the model accurately generates the correct geometry of the level as it passes through the doorway into a new area of the map. In motorway driving, a car is plausibly imagined overtaking on the left. In the supplement, we provide short 16-frame videos generated by each DWM and baseline, as used in the quantitative evaluations. Visual inspection agrees with the ordering of the quantitative metrics; DWM frame-stack > DWM cross-attention ≈ IRIS 64 > IRIS 16 > DreamerV3.

Whilst the above experiments use real sequences of actions from the dataset, we also investigated how robust WM was to novel, user-input actions. Figure 8 shows the effect of the actions in motorway driving – conditioned on the same $H = 6$ real frames, we ask WM to generate trajectories conditioned on five different action sequences. In general the effects are as intended, e.g. steer straight/left/right moves the camera as expected. However, when 'slow down' is input, the model predicts that the traffic ahead has come to a standstill, and the distance to the car in front decreases. But in the real world, slowing down should *increase* the distance to the car in front! Figure 7 shows similar sequences for CS:GO. For the common actions (mouse movements and fire), the effects are as expected, though they are unstable beyond a few frames, since such a sequence of actions is unlikely to have been seen in the demonstration dataset. We note that these issues – the causal confusion and instabilites – are a symptom of training the model on offline data, rather than being an inherent weakness of DWM.

Table 3 provides ablations for DWM. The number of sampling steps, $\mathcal{T}$, allows a trade-off between visual quality and sampling speed – Figure 6 visualizes trajectories for various values of $\mathcal{T}$. Employing CFG for action conditioning was not helpful – even at low weight values, $w > 1.0$, and with static thresholding, it led to over-saturated images that interacted badly with autoregressive sampling.

## 5.2 REINFORCEMENT LEARNING IN ATARI ENVIRONMENTS

### 5.2.1 ENVIRONMENTS, BASELINES, METRICS

A key application of world models consists in training RL agents with simulated trajectories (Sutton, 1991), thereby greatly reducing the total number of interactions with the actual environment. Atari games (Bellemare et al., 2013) are often used to evaluate the sample-efficiency of RL algorithms (Kaiser et al., 2019; Yarats et al., 2021; Schwarzer et al., 2021; Ye et al., 2021; Micheli et al., 2022; Hafner et al., 2023), where a constraint of only 100k actions per environment is imposed to agents, a 500 fold decrease compared to the usual frames budget for Atari environments (Mnih et al., 2015). In terms of human gameplay, this limitation translates to roughly 2 hours of game time.

In this sample-efficient RL setting, we consider eight environments evaluating a wide range of capabilities: *Asterix, Boxing, Breakout, Kangaroo, Krull, Qbert, RoadRunner, UpNDown*. For baselines, we again use DreamerV3 and IRIS, both developed for the purpose of training RL agents in the imagination of their world models. We note other approaches have reached superior results, such as the model-free BFF (Schwarzer et al., 2023) or the model-based EfficientZero (Ye et al., 2021).

We report the usual metrics for Atari games – reward and the human normalized score, $\frac{score\_agent - score\_random}{score\_human - score\_random}$, where *score_random* corresponds to a random policy, and *score_human* corresponds to human players with two hours of experience (Wang et al., 2016).

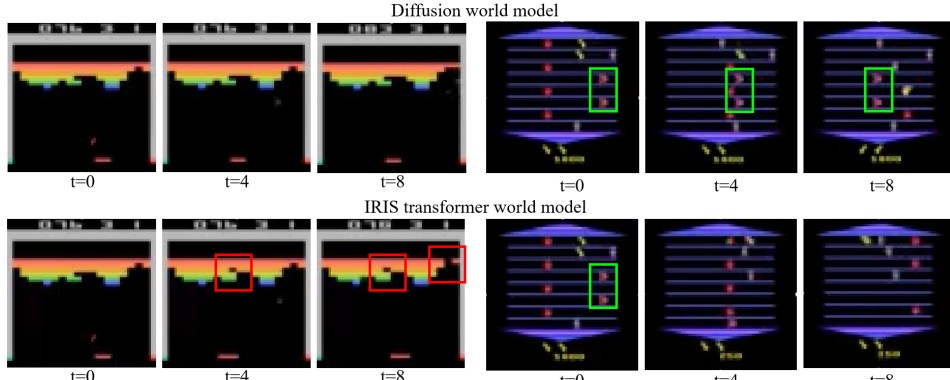

Figure 4: Imagined trajectories in Atari by DWM and IRIS. In *Breakout* (left), while DWM accurately models a single block being broken, IRIS lacks the visual fidelity for this, and makes several mistakes (highlighted in red). In *Asterix* (right), the two red sprites are accurately modeled by DWM (highlighted in green), but erroneously disappear from view with IRIS's prediction. See the supplementary material for playable world models and videos of long-term open-loop generations.

### 5.2.2 RESULTS

Table 2 displays results for DWM frame-stack, averaged over three training runs, each evaluated over 100 episodes following training. We also report published results for IRIS and DreamerV3.

After only two hours of real-time experience, DWM achieves a superhuman score on *Breakout*, *Krull*, and *RoadRunner*, and outperforms both baselines on *Asterix* and *RoadRunner*. DWM bests IRIS on 4/8 games, and it bests DreamerV3 on 3/8 games.

Figure 4 shows trajectories imagined by IRIS and DWM in *Breakout* and *Asterix*. In order to avoid prohibitive sampling speeds, IRIS uses a small number of tokens to encode frames, which results in erroneous predictions. Some bricks are hallucinated or missing in *Breakout* and enemies are mistaken for rewards in *Asterix*. On the other hand, DWM perfectly predicts future frames. Surprisingly, IRIS outperforms DWM in *Breakout*, even though its imagined trajectories are consistently worse than DWM's. We hypothesize that in this game, the ability to accurately predict the movement of the ball is essential and other game dynamics are not as important. On the other hand, disappearing enemies or mistaking enemies for rewards has a dramatic effect on the RL training, which explains the large gap in performance between DWM and IRIS on *Asterix*.

IRIS and DreamerV3 deliver strong results on *Boxing*, but DWM falls short on that particular game. One potential explanation would be that DWM's short-term memory makes it difficult to recreate the movement patterns of the opponent, hence the policy cannot exploit these behaviors.

## 6 CONCLUSION

We proposed DWM, a diffusion-based world model for simulating complex worlds. Through experiments in 3D environments, we showed that DWM provides a visual fidelity superior to existing methods, while maintaining competitive sampling rates. Through experiments in Atari games, we also demonstrated DWM's potential as a surrogate environment for the development of RL agents.

Currently, DWM conditions only on a few previous observations and actions, making it most useful in environments where most of the state information is captured by a short-term memory. Designing an architectural prior capable of handling worlds where long-term memory is paramount could be explored in the future. Another interesting line of research would be to bridge the gap between offline and online datasets. The former provide large amounts of data in real-world settings, while the latter provide state coverage tailored to an RL agent.

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
