# OpenReview forum: "Diffusion World Models"
_ICLR.cc/2024/Conference — Submitted to ICLR 2024_

### Official Review · Reviewer_ozH8 · 2023-10-21

**Soundness:** 3 good
**Presentation:** 3 good
**Contribution:** 2 fair
**Rating:** 5
**Confidence:** 3

**Summary:**

The paper presents a world model architecture for video prediction and control that, compared to previous approaches, adopts a diffusion denoising process for image generation. The work is compared to IRIS and DreamerV3 previous works and several ablations are presented in the main paper and the Appendix.

**Strengths:**

* **Presentation:** I found the work easy to understand as it's well-presented and well-structured. I have some minor remarks that I will present in the Questions section of my review.
* **Motivating problem**: I think the motivation for the work is sound and clear: higher visual fidelity in the world models' predictions, and diffusion-based models should be helpful in addressing this problem
* **Supplementary material**: the additional visualizations and code are appreciated to understand the work better

**Weaknesses:**

* **Novelty**: while there is no work that is alike, there seems to be no major novelty introduced in this paper, as the work mainly grounds on previous approaches in world models (e.g. behaviour learning is performed as in IRIS) and (video) diffusion models.
* **Evaluation in the video prediction task**:  for the video generation experiments, on CS:GO and the driving dataset, the authors only compared to IRIS and DreamerV3, which are not really expected to perform well in this benchmark. I suggest the authors compare their work to more adequate baselines, e.g the ones presented in [1] and [2]. This would help in understanding how effective their method is in action-conditioned and unconditional video generation.
* **Performance in the control task**: when compared to IRIS and DreamerV3 on (a reduced set of) the Atari100k benchmark, the performance of DWM is not really superior. While this is understandable for DreamerV3, as behaviour is learned differently, it is surprising that the method performs worse than IRIS in 4/8 games. It would also be interesting to see what are the performance in the other games of the 100k benchmark, that are currently not shown.

Given the limited technical novelty, I would expect the performance of this approach to be strong in order to recommend acceptance, which doesn't seem to be the case. For this reason, I am leaning towards rejecting the paper.

[1] Temporally Consistent Transformers for Video Generation, Yan et al, 2023

[2] A Control-Centric Benchmark for Video Prediction, Tian et al, 2023

**Questions:**

Suggestions:
* Adding more baselines for the video generation experiments (see weaknesses).
* Providing additional insights into why DWM fails to perform better in certain environments in Atari. (the authors provided an hypothesis for Boxing, which should be easy to test )

Minor remarks:
* the authors state `Hafner et al. (2020b) argue that they need only model task-relevant information, which necessarily
leads to limited image predictions.`, but I could not find such a statement in the paper. I think this statement mostly applies to other lines of work, such as MuZero. I would appreciate if the authors could clarify this or update the citation.
* Q2 and Q3 in the experiments section require trivial YES/NO answers. I would rather turn those into some introductory sentences that explain what is going to be shown in Section 5, rather than having them as questions.

---

### Official Review · Reviewer_cDZi · 2023-10-30

**Soundness:** 3 good
**Presentation:** 3 good
**Contribution:** 2 fair
**Rating:** 5
**Confidence:** 5

**Summary:**

The authors replace the encoder and dynamics model of an IRIS model with a diffusion-based video dynamics model, and compare it to other model-based techniques, such as vanilla IRIS and Dreamerv3.  The diffusion model learns to predict the next pixel observation conditioned on a past window of observations and actions, along with the current action.  Then, it can essentially replace the encoder and latent dynamics models of reconstruction-based model-based reinforcement learning models.  The videos produced were judged in terms of visual quality metrics, as well as how useful they were towards solving RL tasks.

**Strengths:**

The strengths of this work lies in its thorough experimentation on the design choices that enable diffusion models as action and history conditioned dynamics predictors.  The authors find that frame-stacking outperforms cross-attention, and that a particular choice in denoising objective is more robust than classical DDPM.

Originality: This work is decently original.  Although the authors simply replace the IRIS model with a visual dynamics model learned through diffusion, the diffusion modeling itself conditioned on past frames and actions is novel.
Quality: The quality of this work could be improved with more thorough experimentation and well-motivated reasoning.
Clarity: The paper is clear on their work and contributions.
Significance: This paper is not of particularly high significance, in that the results do not substantially outperform existing world models approaches such as Dreamerv3.  Furthermore, DWM only conditions on 6 frames, and may not be a particularly scalable solution (especially when eventually used to model visual environments with larger and larger visual resolution).

**Weaknesses:**

This work proposes utilizing diffusion to learn high-fidelity videos conditioned on past observations and actions.  This then can be used to learn a world model.  However, the motivation behind the approach is not fully motivated.

The authors compare the video generated from their approach according to metrics from video generation literature, such as FID and FVD.  Indeed, it is not particularly surprising that the proposed model works better according to these metrics.  The other models that they compare against are model-based approaches with learned latent embeddings and dynamics.  This makes for an unfair comparison, as such models are not naturally expected to have as high fidelity in terms of image generation as something that operates purely in image space.  The pixel reconstruction objective of MBRL is not necessarily there to learn exact video reconstructions, but as a prior to help learn useful latents which is useful for planning in imagination.

This work begs the question of whether or not super-high visual quality even matters for reinforcement learning; the authors make the assumption that it does, but this reviewer does not believe such a belief should be taken for granted.  Indeed, for the purpose of encoding information, not all the visual details need to matter.  For example, the visual pattern of a floor being checkered or plain is not necessary for learning how to walk over it.  Indeed, fixed patterns, textures, and other visual details often are not relevant to solving a task at hand, and therefore compressing a visual scene into a latent can still be powerful enough to solve the task (where such task-irrelevant visual details can be reconstructed cheaply through a simultaneously learned decoder, allowing the latent to focus on hopefully task-relevant information).  Even as humans, we often are unable to recall in explicit, vivid visual detail what occurred in the past, or what we imagine might happen in the future.  Rough, fuzzy, visual memories and/or plans are often enough for human needs.  This suggests from a huamn-inspired angle that perhaps hyper-resolution planning is not of paramount importance, and generally accurate reconstructions may be good enough for solving tasks (with perhaps more accurate visual reconstructions of things that are relatively more important).  Then, whereas it is understandable that visual quality of predictions should improve, it does not necessarily need to be of utmost important to model in a hyper-detailed manner.  This reviewer believes that this explains why Dreamerv3 does not substantially suffer in the provided Atari results over DWM, despite having worse visual generation fidelity.  Indeed this reviewer believes that the utilized metrics of FID, FVD, etc. may not actually be the right metrics to use for the purposes of understanding usefulness for MBRL efforts, and can be potentially misleading.

Furthermore, a weakness of the work stems from the experiments chosen to showcase DWM.  DWM was motivated as a world model; however, the experiments to showcase the environments it can learn policies for was limited to Atari.  This reviewer is curious as to why only the Atari experiments were performed, when world models have also been applied to continuous control tasks such as the DeepMind Control Suite, Minecraft, DMLab, and more (such as in the listed Dreamerv3).  Furthermore, despite showcasing visual quality results on CSGO and Driving, no policies were learned for them.

To be more thorough in comparing with existing world models, this reviewer suggests comparing against world models that do not rely on reconstruction, such as TDMPC.

Another interesting decision is why offline datasets were considered at all, given that they are prone to causal confusion and instabilities, as the authors stated in the paper.  Indeed, the authors only visualize fidelity of videos for such datasets, which is not necessarily the best metric to care about when looking to learn policies.  The authors do not learn policies from DWM trained on offline datasets.

**Questions:**

1. The authors only utilize a window of 6 frames.  It would be interesting to explore the difference in performance when less or more frames are used, especially for a singular frame for one-step predictions.  It would also be interesting to understand the time complexities of training a DWM with less or more frames (e.g. what is the max conditioning possible).
2. How

---

### Official Review · Reviewer_CvA6 · 2023-10-31

**Soundness:** 3 good
**Presentation:** 2 fair
**Contribution:** 2 fair
**Rating:** 5
**Confidence:** 4

**Summary:**

The authors present "diffusion world models", an approach to world modelling based on video diffusion models. The authors take recent advances in diffusion modelling and augment those techniques with action conditioning, creating a world model. The authors demonstrate the visual ability of their models through various datasets and video metrics, and report performance on a collection of Atari games.

**Strengths:**

- The authors demonstrate video generation that can be conditioned on actions
- Making use of recent advances in diffusion modelling seems like the right path forward for world modelling

**Weaknesses:**

- The authors show advances in terms of video metrics, but do not demonstrate this is a required ingredient for the generation of good world models
- The experiments investigating improvement on downstream RL based on world modelling do not show improvement over existing methods

**Questions:**

I find the paper well written, and nice to read, but in my opinion the work lacks substantial methodological innovation. In the absence of that, I would expect the experiments to be very strong, but neither video metrics nor RL performance provides a very convincing argument. While video metrics are good compared to others, these metrics are agnostic to actions, and hence of limited value in evaluating a world model. I would expect a useful metric to include the faithfulness to some action, and compare that against the baselines. The downstream performance seems limited compared to existing work, and it is not clear it is worth spending the additional computational effort on high quality visual features. Moreover, it is difficult to make a careful assessment in the absence of error bars on the return scores. In my opinion, the most impressive part of the paper is in Figs. 7 and 8, but the results presented there are too few to carefully asses the model capabilities.
Questions:
- Can the authors provide more material like Figs 7 and 8 in the appendix? It is not clear how well the learned model generalizes.
In the absence of substantial improvement in terms of downstream performance, this seems like an important thing to demonstrate.
- Video metrics are nice, but I don't think are as relevant as faithfulness to the input action. I am not aware of such a "conditional video metric", but it would be great if the authors could evaluate their model and their baselines on such a metric. It would be a great opportunity to introduce one that other researchers can use.
- Can uncertainties in Table 2 be added? Interpreting the results without is not possible.
- Can the authors add a quantitative result that demonstrates their world model indeed behaves like a world model? This question is related to the earlier question about the faithfulness to input actions.

---

> ### Author Response · Authors · 2023-11-23
> **Response to Reviewer CvA6**
>
> We thank Reviewer CvA6 for their detailed and insightful review.
>
> **Q1 – Model generalization**
>
> Thank you for highlighting the value of Figure 7 & 8. [Here is a link](https://drive.google.com/drive/folders/1aaCx2TljjKh4nefccktxk286npeRFf7g?usp=sharing) to several videos showing generated trajectories, beginning from the same initial state and applying various constant actions, for CS:GO and motorway driving. Since the world model is trained on offline datasets from expert-level policies, some of the trajectories quickly become out of distribution when certain constant action sequences are input. For example, the driving dataset does not contain trajectories where a steering is applied to take the car off the road. Hence, whilst the world model tends to do well at modeling immediate frames, they sometimes do not generalize well when pushed beyond this. We note this is a symptom of the training data, rather than anything specific to our diffusion world model.
>
> **Q2 – Metrics testing faithfulness to actions**
>
> As the reviewer notes, a perfect action-conditioned video metric is hard to find. However, the LPIPS metric is, to the best of our knowledge, the current favored choice. LPIPS compares frames in a pairwise fashion (real to generated), and in order to do well in this metric, the effect of an action _must_ be modeled correctly. Note however that it does not account for an environment’s stochasticity, hence being imperfect. On the other hand, the FVD metric measures action faithfulness less directly, but _does_ account for stochasticity in the environment being modeled. Hence when combined, the two metrics are able to measure action faithfulness and stochasticity.
>
> **Q3 – Table 2 uncertainties**
>
> We are in the process of rerunning this benchmark, but have not been able to complete this during the rebuttal period. We will share a fully updated table in due course.
>
> **Q4 – Quantitative metrics demonstrating world model behavior**
>
> Thank you for this question, which we believe we were not clear enough about in the main paper. The purpose of dividing the experiments into two sections was to show: 1) DWM can be effective as part of a typical dyna-style algorithm to train an RL agent in imagination, which is a holistic test of DWM’s abilities. 2) DWM also scales to visually complex environments. We chose to demonstrate this in real-world motorway driving and a modern video game. Due to the complexity of these environments, however, we were unable to also test dyna-style agents in these settings, and hence report the best available offline metrics.

---

### Official Review · Reviewer_8zjn · 2023-10-31

**Soundness:** 2 fair
**Presentation:** 3 good
**Contribution:** 2 fair
**Rating:** 3
**Confidence:** 3

**Summary:**

This paper delves into the concept of a world model for reinforcement learning. The authors propose utilizing a diffusion model to create a world model, which can then be harnessed by a reinforcement learning agent. The authors' primary innovation centers around the parametrization and implementation of the diffusion model for a reinforcement learning setting.

To assess the effectiveness of their approach, the authors conducted experiments in various environments with varying levels of visual complexity. For instance, they tested their model generation capability on environments such as CS:GO which offer intricate 3D visual worlds  and real-life video inputs from the Motorway Driving dataset,

In addition to exploring the model's capability to generate videos in the CS:GO and Motorway datasets, the authors also report the performance of their reinforcement learning-trained agent in the context of the Atari benchmarks.

**Strengths:**

This paper investigates the utilization of a diffusion model as a world model, which appears novel to me a. The authors also put forward an empirical assessment that encompasses both visually intricate tasks, such as CS:GO and Motorway, and more synthetic benchmarks like the Atari games

**Weaknesses:**

One of the central hypotheses presented in this paper underscores the significance of enhancing pixel-level predictions to develop a task-agnostic world model, which can then be harnessed by a general RL agent. However, this hypothesis remains unverified, as the authors do not assess the performance of an RL agent in the context of CS:GO and Motorway driver tasks; instead, they solely focus on evaluating video generation capabilities. It remains uncertain why improved video generation metrics would necessarily correlate with improved performance in downstream RL tasks, which is the ultimate objective of a world model.

Furthermore,  it is unclear from the empirical study concerning the Atari environment that a world diffusion model offers advantages over alternatives like Dreamerv3 and Iris in terms of returns or human-normalized scores?

**Questions:**

Expanding on my comment in the Weaknesses section, I'm curious why the authors chose not to train an RL agent on the CS:GO and Motorway tasks.  More generally, I believe it is crucial to provide evidence, possibly through an other reinforcement learning tasks, that a superior world model, as indicated by better generation quality, indeed corresponds to improved performance in downstream tasks. This validation would underscore the significance of the proposed approach.

---

### Author Response · Authors · 2023-11-22
**General response**

Dear ICLR Reviewers,

We sincerely appreciate the time and effort you dedicated to reviewing our manuscript. We are grateful to you for highlighting positive aspects of the paper, reaffirming that our work is in a direction of interest to the community, as well as your constructive comments guiding us on how to improve the paper. In this note we summarize these positive aspects as well as the main criticisms, alongside brief responses.

**Positive Highlights:**

- **Reviewer ozH8:** Recognized the paper's strengths in being well-presented, with a sound and clear motivation, and supported by good supplementary material.
- **Reviewer CvA6:** Commended the demonstration of action-conditioned video generation, the rationale for using diffusion for world modeling, and the overall clarity of the manuscript.
- **Reviewer cDZi:** Appreciated the ablations on design choices, originality, and clarity of presentation.
- **Reviewer 8zjn:** Acknowledged the novelty and the evaluation in both visually intricate tasks (CS:GO/Motorway) and synthetic games (Atari).

**Response to main feedback:**

1. **Atari experiments.** We agree with the reviewers' consensus that stronger experiments in Atari would further enhance the paper's contributions. For our current results we performed only minimal tuning of the RL algorithm – we saw our contribution as the world model rather than the RL portion - and are encouraged that DWM already outperforms IRIS and DreamerV3 on some of the games considered. Nevertheless, work is underway to increase the number of Atari environments tested and conduct analysis to understand when DWM performs well and when it underperforms. We have unfortunately not been able to conclude this work during this short rebuttal period.

2. **Does visual quality matter? (Reviewers 8zjn, cDZi):** Thank you for your insights on this. On reflection we believe that the paper would benefit from a broader discussion of the importance of prediction quality in world modeling, and will include several of the points raised by reviewers. However, we do stand by the belief that better visual quality is often (though admittedly not always) valuable, with task-agnostic world modeling as one example.

3. **Unfair comparison to IRIS and Dreamer for video generation (Reviewers cDZi, ozH8):** We acknowledge that the primary design goal of Dreamer was not to learn high-quality generations, and it serves as a weak baseline in this respect. Nevertheless we felt it would be of interest to readers to include as a point of reference. On the other hand, IRIS is exclusively trained on a reconstruction objective (split across two models), and forms what is (to our knowledge) the strongest world model in this respect. Whilst other models may also perform well in this respect (e.g. TECO), they have not been proposed as world models, and we considered it out-of-scope to evaluate them here.

4. **Evaluation in Motorway and CS:GO (Reviewers 8zjn, cDZi, CvA6):** We agree that the video quality metrics only test one aspect of the world model. However, going beyond these and running RL experiments in Motorway and CS:GO are challenging for several reasons. 1) There is no clear reward function in driving. 2) Rolling out driving policies in the real world is infeasible. 3) Whilst CS:GO is a simulated environment, it is not available as a standard RL environment and allows only a limited form of online interaction. It was because of these challenges that we included experiments on Atari where rewards can be optimized straightforwardly.

Once again, we express our gratitude for your time and expertise.

Warm regards,

The authors

---

> ### Comment · Reviewer_CvA6 · 2023-11-22
>
> While I appreciate a high level response like this, I do feel as if the questions I posed to the authors are not actually answered. Could the authors respond to my original questions?
>
> The response under 4. in the general response only tangentially addresses my concerns, and does not answer any of the questions I asked.

---

> > ### Author Response · Authors · 2023-11-23
> >
> > Thank you for the prompt. We respond to your queries in a direct response below.

---

### Meta-Review · Area_Chair_YnUt · 2023-12-09

**Metareview:**

**Summary**

The paper presents "diffusion world model" in reinforcement learning, a novel adaptation of diffusion models for video generation applied to world modeling. This method, tested in diverse environments like CS:GO and the Motorway Driving dataset, shows promising results in generating complex visual worlds and excels in various Atari benchmark games. The model's innovation lies in its ability to effectively predict pixel observations and replace traditional components in model-based reinforcement learning frameworks, as evidenced by comparisons with established models like IRIS and DreamerV3.

**Strengths**

1. The paper introduces a novel use of diffusion models in world modeling for reinforcement learning.

2. It features extensive empirical testing across diverse environments, from complex 3D visuals to Atari games, demonstrating the model's versatility.

3. The work is notable for its thorough examination of design choices, particularly in action and history-conditioned dynamics prediction using diffusion models.

4. The paper is well-structured and clear, supplemented with helpful visualizations and code, enhancing understanding and practical application.


**Weaknesses**

While the paper proposes interesting concept of using diffusion models as the world models, the reviewers raised several concerns:

1. The central hypothesis is unverified. The authors did not assess the RL agent's performance in complex tasks like CS:GO and Motorway, focusing instead only on video generation capabilities. Even though the rewards are not easy to define, it is still needed to show the real RL improvement.

2. The paper is also criticized for lacking substantial methodological innovation. The experiments conducted, including video metrics and RL performance, are not sufficiently convincing to support the proposed approach. The utility of improved video metrics in evaluating a world model's effectiveness is questioned, particularly in the context of action conditioning.

3. The motivation behind using high-fidelity video predictions for reinforcement learning is not fully justified. Comparing the model to others that are not expected to perform as well in image generation (like IRIS and DreamerV3) is seen as unfair. It seems to be an open question whether super-high-quality video is essential or not.

4. The experiments are limited primarily to the Atari environment, with no policies learned for more complex or continuous control tasks. The choice of metrics like FID and FVD may not be appropriate for assessing the model's utility in model-based reinforcement learning. Additionally, there's criticism regarding the lack of comprehensive performance evaluation across various tasks and environments, and a failure to compare the method against non-reconstruction-based models like TD-MPC.

**Justification For Why Not Higher Score:**

Overall, all the reviewers maintained a negative perspective towards accepting the paper and indicate the bigger room for further improvement with additional justification on the motivation and more traditional RL environments.

**Justification For Why Not Lower Score:**

N/A

---

### Decision · Program_Chairs · 2024-01-16

Reject